# Changes over Time in Hemoglobin A1C (HbA_1C_) Levels Predict Long-Term Survival Following Acute Myocardial Infarction among Patients with Diabetes Mellitus

**DOI:** 10.3390/jcm10153232

**Published:** 2021-07-22

**Authors:** Ygal Plakht, Harel Gilutz, Arthur Shiyovich

**Affiliations:** 1Department of Nursing, Faculty of Health Sciences, Ben-Gurion University of the Negev, Beer Sheva 8410501, Israel; 2Department of Emergency Medicine, Soroka University Medical Center, Beer Sheva 84101, Israel; 3Goldman Medical School, Faculty of Health Sciences, Ben-Gurion University of the Negev, Beer Sheva 8410501, Israel; Gilutz@bgu.ac.il; 4Department of Cardiology, Rabin Medical Center, Petah Tikva, Israel and Sackler Faculty of Medicine, Tel Aviv University, Tel Aviv 6997801, Israel; arthur.shiyovich@gmail.com

**Keywords:** diabetes mellitus, acute myocardial infarction, hemoglobin A1c, prognosis, mortality

## Abstract

Frequent fluctuations of hemoglobin A1c (HbA_1C_) values predict patient outcomes. However, data regarding prognoses depending on the long-term changes in HbA_1C_ among patients after acute myocardial infarction (AMI) are scarce. We evaluated the prognostic significance of HbA_1C_ levels and changes among diabetic patients (*n* = 4066) after non-fatal AMI. All the results of HbA_1C_ tests up to the 10-year follow-up were obtained. The changes (∆) of HbA_1C_ were calculated in each patient. The time intervals of ∆HbA_1C_ values were classified as rapid (<one year) and slow (≥one year) changes. The outcome was all-cause mortality. The highest mortality rates of 53.8% and 35.5% were found in the HbA_1C_ < 5.5–7% and ∆HbA_1C_ = −2.5–(−2%) categories. A U-shaped association was observed between HbA_1C_ and mortality: adjOR = 1.887 and adjOR = 1.302 for HbA_1C_ < 5.5% and ≥8.0%, respectively, as compared with 5.5–6.5% (*p* < 0.001). Additionally, ∆HbA_1C_ was associated with the outcome (U-shaped): adjOR = 2.376 and adjOR = 1.340 for the groups of <−2.5% and ≥2.5% ∆HbA_1C_, respectively, as compared to minimal ∆HbA_1C_ (±0.5%) (*p* < 0.001). A rapid increase in HbA_1C_ (but not decrease) was associated with a greater risk of mortality. HbA_1C_ values and their changes are significant prognostic markers for long-term mortality among AMI-DM patients. ∆HbA_1C_ and its timing, in addition to absolute HbA_1C_ values, should be monitored.

## 1. Introduction

The global burden of diabetes mellitus (DM), a major cardiovascular risk factor, has increased dramatically throughout the last two decades [1,2]. Acute myocardial infarction (AMI) is a leading cause of increased mortality among patients with DM [3,4,5,6,7]. Furthermore, DM is a significant risk factor for short- and long-term morbidity and mortality among patients presenting with AMI [8,9]. Glucose and glycated hemoglobin (HbA_1C_) levels upon admission with AMI were associated with long-term mortality among patients with and without DM [10]. Thus, one of the cornerstones of treatment is glycemic monitoring and control, usually monitored by serial HbA_1C_ levels [11], based on evidence from several trials reporting improved outcomes with lowering of HbA_1C_ levels [4,12,13].

Recently, glycemic variability emerged as an additional and possibly even superior predictor of diabetic complications than mean HbA_1C_, with frequent fluctuations independently associated with poor prognosis. In contrast, more stable HbA_1C_ control may benefit patients [14,15,16,17,18,19,20,21]. In addition, Lee et al. recently reported that long-term mean HbA_1C_ was significantly associated with long-term survival following AMI [22]. However, data regarding long-term HbA_1C_ changes following AMI and the association with the survival of these patients are scarce. Thus, this study aimed to evaluate the prognostic significance of HbA_1C_ and its changes among patients with DM with non-fatal AMI.

## 2. Materials and Methods

### 2.1. Study Population

This observational study retrospectively evaluated consecutive patients with DM admitted with a non-fatal AMI at Soroka University Medical Center (SUMC), a tertiary medical center in Southern Israel, from 1 January 2002 through 31 December 2017, who survived at least one year post-discharge. Patients were excluded if they had the following criteria: under age 18 years or no HbA_1C_ measurements throughout the follow-up period. The SUMC ethics committee approved the study (SOR-0319-16).

### 2.2. Data Sources and Classifications

The baseline data were obtained from the electronic medical records of SUMC and included: patient demographics, comorbidities, and results of laboratory tests and echocardiographic, angiographic and revascularization procedures, as previously reported for the Soroka Acute Myocardial Infarction (SAMI) project [23]. The diagnosis of AMI was based on the International Classification of Diseases, Ninth Revision, Clinical Modification (ICD-9-CM) codes: ST-segment elevation AMI (STEMI) 410.0 *–410.6 * and non-ST-segment elevation AMI (NSTEMI) 410.7 *–410.9 *. The DM definitions (and characteristics) were as follows: ICD-9-CM 250 * and/or HbA_1C_ ≥ 6.5% throughout six months before or after hospitalization. The diagnosis of anemia was based on low hemoglobin blood levels (for males < 13 g/dL; for females < 12 g/dL) during the index hospitalization. Patients were defined as having significant renal failure if they were either on hemodialysis or had an estimated glomerular filtration rate (eGFR) < 30 mL/min/1.73 m^2^. Either the ICD-9-CM code defined dyslipidemia, or this was given if the low-density lipoproteins (LDLs) were ≥100 mg/dL. Severe left ventricular (LV) dysfunction was defined as an ejection fraction < 30%.

### 2.3. Follow-Up and the Study Endpoint

Follow-up started after discharge and continued up to 10 years after that, or until 1 July 2020. The outcome was all-cause mortality. The survival status or death date for each patient was obtained from the Ministry of the Interior’s population registry.

### 2.4. HbA_1C_ Values and Changes during the Follow-Up

All the HbA_1C_ values during the follow-up were obtained and classified as follows: <5.5%, 5.5–6.5%, 6.5–7.0%, 7.0–8.0% and ≥8.0%. In addition, the changes (differences between every pair [later result–earlier result] of HbA_1C_ values; ∆HbA_1C_) throughout the follow-up of each patient were calculated and classified as follows: <−2.5%, −2.5–(−2.0)%, −2.0–(−1.5)%, −1.5–(−1.0)%, −1–(−0.5)%, −0.5–0.5%, 0.5–1%, 1.0–1.5%, 1.5–2.0%, 2.0–2.5% and ≥2.5%. The time interval of ∆HbA_1C_ change values was classified as rapid (<one year) or slow (≥one year).

### 2.5. Statistical Analysis

Statistical analysis was performed using IBM SPSS 26 (SPSS Inc., Chicago, IL, USA) software. Patient characteristics were presented as mean and standard deviation (SD) for continuous variables and numbers (*n*) and as percentages (%) for the categorical data. A visual inspection of the continuous variables showed that their distribution was “normal”. Comparison of the baseline characteristics and the outcomes was performed using the chi-square test, the chi-square test for linear trend of categorical variables and Student’s t-test for continuous variables. A parametric test was used based on the central limit theorem and large sample size. The measure of association between HbA_1C_ levels during the follow-up and the risk of mortality was assessed using generalized estimating equation (GEE), binary logistic type, which considered the repeated measures of HbA_1C_ for each patient. Three different models were built: (1) HbA_1C_ levels alone, (2) ∆HbA_1C_ levels alone and (3) HbA_1C_ and ∆HbA_1C_ levels, with the time from hospital discharge and the baseline characteristics as the potential confounders, in order to assess the disparity of the associations between ∆HbA_1C_ values and outcomes by the groups of time intervals of ∆HbA_1C_ (rapid vs. slow change). This analysis included a multivariate model (GEE), which included the interactive variable of ∆HbA_1C_ by time interval. The reference categories for these models were: HbA_1C_ 5.5–6.5% and the minimal category of HbA_1C_ change (−0.5–0.5%). The results of the models are presented as adjusted odds ratios (adjORs) and 95% confidence intervals (CIs) for ORs. For each test, *p* < 0.05 was considered statistically significant.

## 3. Results

A total of 4066 eligible DM patients were analyzed in the current study, with a mean age of 66.4 ± 11.9 years; 36% were females. The median follow-up (after the first year) was 6.1 years, with 1802 (44%) patients dying throughout the follow-up period. The baseline characteristics of the study cohort and comparisons (at the index date) between patients who survived and those that died are presented in Table 1. Patients who died were older and more likely to be women, with higher rates of prior AMI, prior coronary artery bypass graft (CABG), congestive heart failure, renal failure and peripheral arterial disease than the survivors. Furthermore, they had an increased rate of DM-associated target-organ complications compared with the survivors. However, these patients had a lower rate of dyslipidemia, obesity, smoking and family history of early coronary artery disease (CAD) than the survivors. Besides, patients who died had a higher prevalence of non-cardiovascular comorbidity and were more likely to present with NSTEMI (rather than STEMI). Those who died had severe left ventricular dysfunction and were less likely to undergo revascularization due to the index AMI compared to the survivors.

A total of 35,609 HbA_1C_ test results were documented among the study cohort during the follow-up. HbA_1C_ values ranged between 3.7% and 18.6%, with a mean of 7.81 ± 1.74%. The mean HbA_1C_ was lower among patients who died vs. survivors (7.75 ± 1.82% vs. 7.84 ± 1.69%, respectively, *p* < 0.001). The mean values of HbA_1C_ throughout the follow-up (by follow-up year) among patients who died and among survivors are presented in Figure A1 (Appendix A). A trend of decrease in HbA_1C_ was observed among patients who died, while values remained similar or even increased over time among survivors; changes were most evident after five years of follow-up.

∆HbA_1C_ was calculated between 229,645 pairs of HbA_1C_ values. The time between HbA_1C_ tests ranged between 90 and 3645 days (median of 863 days [interquartile range 420–1518]). HbA_1C_ values decreased among patients who died and increased among survivors (−0.09 ± 1.65% vs. 0.10 ± 1.53%, respectively, *p* < 0.001).

The mortality rates and the relative risks according to the different HbA_1C_ and ∆HbA_1C_ categories are presented in Figure 1. Significant differences in mortality rates between HbA_1C_ and ∆HbA_1C_ levels were demonstrated (*p* < 0.001 for each). The highest death rate was observed in the HbA_1C_ category of <5.5% (53.8%) with an adjOR of 1.852 (95% CI: 1.530–2.242, *p* < 0.001), as compared to the reference group (5.5–6.5%; Figure 1a). However, higher HbA_1C_ values were associated with lower mortality risk: adjORs of 0.798 (95% CI: 0.716–0.890, *p* < 0.001) and 0.825 (95% CI: 0.721–0.943, *p* = 0.005) for the HbA_1C_ categories of 6.5–7.0 and 7.0–8.0 were similar to the category of ≥8.0% (adjOR = 0.891; 95%CI: 0.765–1.037, *p* = 0.136), as compared with the reference category.

Regarding ∆HbA_1C_ (Figure 1b), the risk of mortality increased with negative ∆HbA_1C_ and remained similar with positive ∆HbA_1C_. A decrease of >2.5% in HbA_1C_ was associated with a higher risk of mortality, as compared with the group with the minimal change (±0.5%): 35.3% vs. 26.7%, adjOR 1.495 (95%CI: 1.355–1.650, *p* < 0.001).

Following multivariable adjustment for potential confounders, the association of HbA_1C_ and mortality presented a U-shape-like association, with increasing risk with both higher and lower HbA_1C_ (more prominent; Figure 2a). A similar trend was found between ∆HbA_1C_ and mortality (Figure 2b).

The mortality rates according to the HbA_1C_ changes in the rapid- and slow-changing groups are presented in Figure A2 (Appendix A). Figure 3 displays the adjORs for mortality according to ∆HbA_1C_ in the rapid- and slow-changing groups (adjusted to baseline characteristics, HbA_1C_ levels and time lag from hospital discharge). It is evident that when HbA_1C_ increases rapidly, it carries a greater risk of mortality than a slow increase. However, the rate of decreasing HbA_1C_ has the same risk of mortality irrespective of the time of change.

## 4. Discussion

The current study evaluated the prognostic significance of the levels of and changes in HbA_1C_ in patients with DM following non-fatal AMI. The main findings include: (1) the adjusted risk of mortality increases at HbA_1C_ values ≤ 5.5% and ≥7%; (2) ∆HbA_1C_ is a significant independent prognostic marker for the risk of mortality following AMI with a U-shaped association (stronger association with mortality for a decrease in HbA_1C_); (3) rapid versus slower increase in HbA_1C_ following AMI is associated with a greater risk of mortality. However, the time interval of decrease in HbA_1C_ does not seem to be a significant determinant of mortality risk.

The finding that HbA_1C_ levels are associated with long-term mortality of AMI patients is overall consistent with previous studies of AMI patients [19,24,25]. Other studies often used admission HbA_1C_ values while we evaluated all values (not necessarily upon admission). The association between low and high HbA_1C_ and ∆HbA_1C_ with long-term mortality following AMI is consistent with a recent study by Lee et al. [22] that evaluated the association of long-term mean HbA_1C_ with long-term survival following AMI. They reported that patients with a mean HbA_1C_ value ≥ 6.0% to <7.5% had a better prognosis than patients with HbA_1C_ values of <6.0% or >7.5%. However, in our study, we did not use mean values and quantified the change and the trajectory of the change more elaborately, which is more informative.

Our findings are also consistent, in some parts, with those of Olsson et al. [5] who evaluated HbA_1C_ as an AMI predictor in a large population of patients with type-2 DM and compared three distinct methods using HbA_1C_ measurements taken from diabetes diagnosis and beyond (baseline-, latest- and mean-HbA_1C_). The authors found that the baseline HbA_1C_ variable was a much weaker predictor of AMI than the updated latest and updated mean HbA_1C_ variables, which in our study, are equivalent to the ∆HbA_1C_. In addition, they reported a J-shaped association between the updated latest HbA_1C_ variable and AMI with increased risk associated with HbA_1C_ < 6.0% and >7.0% (reference 6.0–7.0%). Furthermore, a recent secondary analysis from the Action to Control Cardiovascular Risk in Diabetes (ACCORD) trial [26] showed that substantial changes in HbA_1C_ were significantly associated with a higher risk of heart failure: hazard ratio (HR) 1.32 (95% CI: 1.08–1.75) for ≥10% decrease in HbA_1C_ and 1.55 (95% CI: 1.19–2.04) for ≥10% increase in HbA_1C_ (reference < 10% change in HbA_1C_). Among ambulatory DM patients, high HbA_1C_ variability was associated with increased risk of all-cause and cardiovascular mortality and DM complications [21].

Interestingly, Carson et al. [27] reported a U-shaped association with HbA_1C_ levels, more robust in the lower categories (like our study), associated with increased all-cause mortality among US adults without diabetes. The reasons for the increased risk of mortality associated with low levels of HbA_1C_ are unknown. However, it has been suggested that patients in the lowest category (HbA_1C_ < 6.0%) may represent a frail and vulnerable group. Lower glycemic levels can be attributed not only to intensive glycemic control. Other factors such as old age and comorbidities may cause low HbA_1C_ [22,27,28,29,30]. In addition, increased oxidative stress and endothelial dysfunction, which are markers of worse outcomes, were reported to be associated with significant glycemic changes [31,32]. Furthermore, a confounding—rather than causal—relationship may explain the observed association between HbA_1C_ changes and reduced long-term survival. These include poor compliance with medications, lifestyle recommendations, comorbidity, lack of support and infections [15,33,34,35,36].

Nevertheless, Lee et al. [21] recently found an association between hypoglycemic frequency, HbA_1C_ variability and mortality. They concluded that these associations suggest that intermittent hypoglycemia may result in poorer outcomes in diabetic patients. Interestingly, we found that rapid versus slow increase in HbA_1C_ following AMI is associated with greater risk of mortality. Nevertheless, when a decrease in HbA_1C_ occurred, its rate had a minimal impact. This could be explained by an association between the rate of changes in HbA_1C_ and the magnitude of the deleterious consequences of DM (and its various complications) on patients following AMI. Furthermore, it is possible that a rapid increase in HbA_1C_ results from a systemic illness (e.g., infection or inflammation) or a stressor that activates counterregulatory hormones and additional mechanisms that support insulin intolerance and hyperglycemia, which are significant and long enough to cause a rapid increase in HbA_1C_ [37]. Additionally, various medications may signal more complicated cardiac disease (e.g., hydrochlorothiazides, beta-blockers) or systemic illness (e.g., corticosteroids) and may cause an increase in glucose levels, explaining this association [38]. Finally, a decrease of HbA_1C_ may occur due to other causes such as hypoglycemia (often associated with older age and other comorbidities), frailty and comorbidities (e.g., chronic inflammation, liver function derangement), rather than related to DM itself, thus potentially explaining the relatively weak association between the rate of decrease in HbA_1C_ and mortality [21,22,28]. Moreover, changes in HbA_1C_ levels could result from false decreases/increases, occurring due to various clinical situations and diseases such as iron deficiency anemia, deficiency of vitamin B12 or folic acid, severe hypertriglyceridemia, severe hyperbilirubinemia, chronic salicylate ingestion, chronic opioid ingestion, lead poisoning, acute or chronic blood loss, splenomegaly, pregnancy, vitamin E ingestion, red blood cell transfusion, ribavirin and interferon-alpha use, hemoglobin variants, vitamin C ingestion, uremia, hemodialysis and erythropoietin injections [39]. Although such clinical situations and diseases could, on the one hand, bias our findings, they could on the other hand (mostly the diseases) explain the observed increased mortality associated with such fluctuations.

### Limitations

First, this is a non-randomized retrospective observational study; hence, it has the limitations of such a design and is subject to bias; mainly, it cannot show causality. Second, data regarding therapeutic interventions in general, and particularly some medications for DM treatment, were not available and could be a source of bias, particularly since it is currently known that the way in which glucose levels are reduced has a significant effect on the ultimate outcome of diabetic patients [40]. Third, the HbA_1C_ tests were taken at the discretion of the treating physicians rather than at prespecified intervals. Fourth, we used all-cause mortality rather than cause-specific mortality (e.g., cardiovascular). Fifth, data regarding multiple factors that could bias HbA_1C_ levels were not included.

## 5. Conclusions

The study shows for the first time that among DM patients with non-fatal AMI, long-term changes in HbA_1C_ values and the time interval for these changes (when increasing) are significant independent prognostic markers for long-term all-cause mortality with a U-shaped association. Thus, ∆HbA_1C_, in addition to absolute HbA_1C_ values, should be monitored in post-AMI patients. Furthermore, additional studies to further evaluate the mechanisms for these changes and potential targets for interventions are warranted.

## Figures and Tables

**Figure 1 jcm-10-03232-f001:**
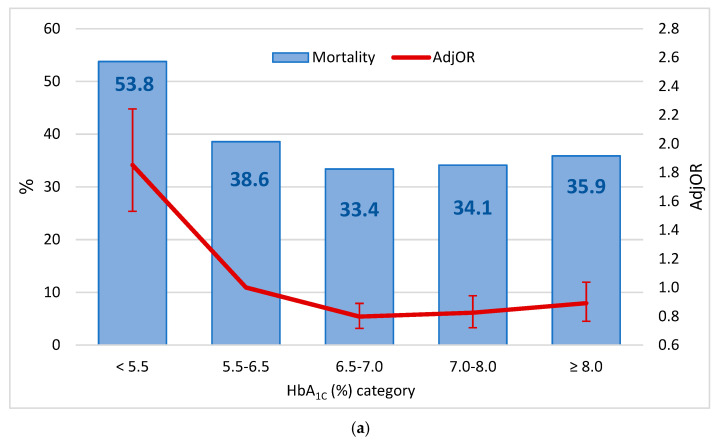
(**a**) The rates and the relative risks of mortality by HbA_1C_ (adjusted odds ratios and 95% CI) categories; (**b**) The rates and the relative risks of mortality (adjusted odds ratios and 95% confidence intervals) by HbA_1C_ categories. (**a**,**b**) are the results of separate models. Abbreviations: adjOR—adjusted odds ratio (adjusted for repeated measures), CI—confidence interval, HbA_1C_—hemoglobin A1C, ∆HbA_1C_—the difference between two HbA_1C_ tests (later–earlier).

**Figure 2 jcm-10-03232-f002:**
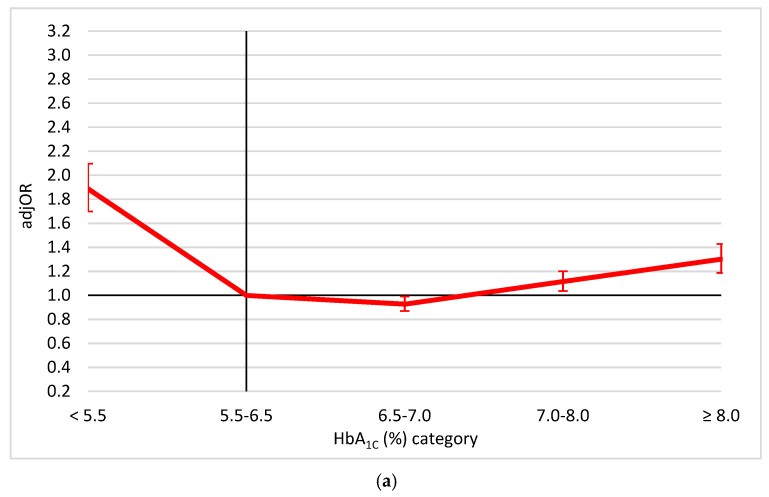
The associations (adjusted odds ratios and 95% confidence intervals): (**a**) Between HbA_1C_ and mortality; (**b**) Between HbA_1C_ change (∆HbA_1C_) and mortality, adjusted for the potential confounders. The results of the combined model included HbA_1C_ levels and ∆HbA_1C_, adjusted for: repeated measures, time from hospital discharge, age, sex, supraventricular arrhythmias, congestive heart failure, chronic ischemic heart disease, history of myocardial infarction, history of percutaneous coronary intervention, history of a coronary artery bypass graft, atrioventricular block, renal failure, diabetes mellitus, hypertension, peripheral vascular disease, chronic obstructive pulmonary disease, neurological disorders, malignancy, anemia, gastrointestinal bleeding, alcohol/drug addiction, type of acute myocardial infarction, severe left ventricular dysfunction, left ventricular hypertrophy and three-vessel/left-main coronary arteries disease. Abbreviations: AdjOR—adjusted odds ratio, HbA_1C_—hemoglobin A_1C_, HbA_1C_ change (∆HbA_1C_)—the difference between two HbA_1C_ tests (later–earlier).

**Figure 3 jcm-10-03232-f003:**
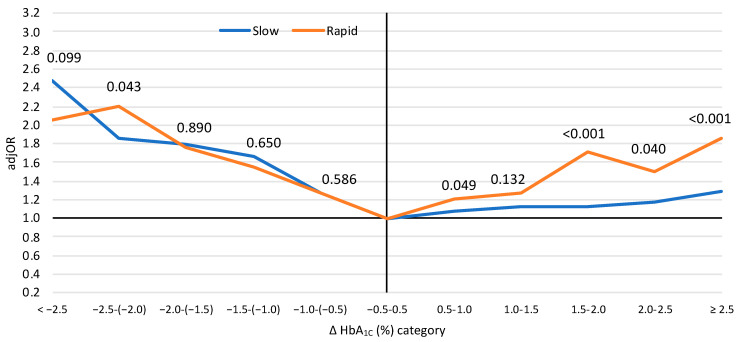
The relative risk (adjusted odds ratios) for mortality according to HbA_1C_ changes in the rapid- and slow-changing groups. Rapid change—time interval of ∆HbA_1C_ values < one year; slow change—time interval of ∆HbA_1C_ values ≥ one year. The numbers represent the *p*-values for disparity (*p* for interaction) between the rapid- and slow-changing HbA_1C_ (separately for each ∆HbA_1C_ category). Adjusted for: repeated measures, HbA1C levels, time from hospital discharge, age, sex, supraventricular arrhythmias, congestive heart failure, chronic ischemic heart disease, history of myocardial infarction, history of percutaneous coronary intervention, history of a coronary artery bypass graft, atrioventricular block, renal failure, diabetes mellitus, hypertension, peripheral vascular disease, chronic obstructive pulmonary disease, neurological disorders, malignancy, anemia, gastrointestinal bleeding, alcohol/drug addiction, type of acute myocardial infarction, severe left ventricular dysfunction, left ventricular hypertrophy and three-vessel/left-main coronary arteries disease. Abbreviations: AdjOR—adjusted odds ratio, HbA_1C_—hemoglobin A_1C_, HbA_1C_ change (∆HbA_1C_)—the difference between two HbA_1C_ tests (later–earlier).

**Table 1 jcm-10-03232-t001:** Baseline characteristics and comparisons between patients who died and those who survived.

Group	Survived	Died	Total	*p*
*n*	2264	1802	4066	
Demographics				
Age, years:				
Mean (SD)	62.18 (11.13)	71.78 (10.61)	66.44 (11.90)	<0.001
<65	1401 (61.9)	457 (25.4)	1858 (45.7)	<0.001
65–75	568 (25.1)	638 (35.4)	1206 (29.7)	
≥75	295 (13.0)	707 (39.2)	1002 (24.6)	
Sex: Male	1613 (71.2)	978 (54.3)	2591 (63.7)	<0.001
Ethnicity: Arab/other	579 (25.6)	265 (14.7)	844 (20.8)	<0.001
Cardiac diseases				
Cardiomegaly	164 (7.2)	206 (11.4)	370 (9.1)	<0.001
Supraventricular arrhythmias	230 (10.2)	398 (22.1)	628 (15.4)	<0.001
CHF	280 (12.4)	542 (30.1)	822 (20.2)	<0.001
Pulmonary heart disease	125 (5.5)	261 (14.5)	386 (9.5)	<0.001
CIHD	2008 (88.7)	1336 (74.1)	3344 (82.2)	<0.001
s/p MI	201 (8.9)	290 (16.1)	491 (12.1)	<0.001
s/p PCI	325 (14.4)	276 (15.3)	601 (14.8)	0.391
s/p CABG	195 (8.6)	276 (15.3)	471 (11.6)	<0.001
AV block	81 (3.6)	99 (5.5)	180 (4.4)	0.003
Cardiovascular risk factors				
Renal diseases	96 (4.2)	350 (19.4)	446 (11.0)	<0.001
Dyslipidemia	2021 (89.3)	1516 (84.1)	3537 (87.0)	<0.001
Hypertension	1468 (64.8)	1122 (62.3)	2590 (63.7)	0.09
Obesity	726 (32.1)	418 (23.2)	1144 (28.1)	<0.001
Smoking	1020 (45.1)	429 (23.8)	1449 (35.6)	<0.001
PVD	219 (9.7)	385 (21.4)	604 (14.9)	<0.001
Family history of IHD	213 (9.4)	33 (1.8)	246 (6.1)	<0.001
Other disorders				
COPD	125 (5.5)	202 (11.2)	327 (8.0)	<0.001
Neurological disorders	310 (13.7)	490 (27.2)	800 (19.7)	<0.001
Malignancy	36 (1.6)	97 (5.4)	133 (3.3)	<0.001
Anemia	892 (39.4)	1148 (63.7)	2040 (50.2)	<0.001
GI bleeding	26 (1.1)	30 (1.7)	56 (1.4)	0.16
Schizophrenia/Psychosis	16 (0.7)	40 (2.2)	56 (1.4)	<0.001
Alcohol/drug addiction	32 (1.4)	23 (1.3)	55 (1.4)	0.707
History of malignancy	91 (4.0)	122 (6.8)	213 (5.2)	<0.001
Characteristics of diabetes mellitus				
Type I	18 (0.8)	21 (1.2)	39 (1.0)	0.229
Insulin-treated	259 (11.4)	185 (10.3)	444 (10.9)	0.233
Complications:				
Non-complicated	1923 (84.9)	1338 (74.3)	3261 (80.2)	<0.001
Renal	121 (5.3)	171 (9.5)	292 (7.2)	<0.001
Peripheral circulation	137 (6.1)	271 (15.0)	408 (10.0)	<0.001
Ophthalmic	128 (5.7)	153 (8.5)	281 (6.9)	<0.001
Neurological	91 (4.0)	78 (4.3)	169 (4.2)	0.624
Other	11 (0.5)	13 (0.7)	24 (0.6)	0.330
Results of HbA_1C_ tests (at baseline)				
HbA_1C_ tests performance	2035 (89.9)	1557 (86.4)	3592 (88.3)	0.001
HbA_1C_, %: Mean (SD)	7.63 (1.69)	7.68 (1.78)	7.65 (1.73)	0.456
Clinical characteristics of the hospitalization				
Type of AMI: STEMI	940 (41.5)	564 (31.3)	1504 (37.0)	<0.001
Results of echocardiography				
Echocardiography performance	1948 (86.0)	1313 (72.9)	3261 (80.2)	<0.001
Severe LV dysfunction	166 (8.5)	221 (16.8)	387 (11.9)	<0.001
LV hypertrophy	113 (5.8)	116 (8.8)	229 (7.0)	0.001
Mitral regurgitation	64 (3.3)	135 (10.3)	199 (6.1)	<0.001
Tricuspid regurgitation	24 (1.2)	82 (6.2)	106 (3.3)	<0.001
Pulmonary hypertension	89 (4.6)	192 (14.6)	281 (8.6)	<0.001
Results of angiography				
Angiography performance	1897 (83.8)	1043 (57.9)	2940 (72.3)	<0.001
Measure of CAD:				
None/non-significant	72 (3.8)	39 (3.7)	111 (3.8)	<0.001
One vessel	456 (24.0)	182 (17.4)	638 (21.7)	
Two vessels	543 (28.6)	237 (22.7)	780 (26.5)	
Three vessels/LM	826 (43.5)	585 (56.1)	1411 (48.0)	
Type of treatment:				
Noninvasive	245 (10.8)	725 (40.2)	970 (23.9)	<0.001
PCI	1581 (69.8)	882 (48.9)	2463 (60.6)	
CABG	438 (19.3)	195 (10.8)	633 (15.6)	

The data are presented as *n* (%) unless otherwise stated. Abbreviations: AMI—acute myocardial infarction, AV—Atrioventricular, CABG—coronary artery bypass graft, CAD—coronary arteries disease, CHF—congestive heart failure, CIHD—chronic ischemic heart disease, COPD—chronic obstructive pulmonary disease, HbA_1C_—hemoglobin A1C, IHD—ischemic heart disease, GI—Gastrointestinal, LM—left main (coronary artery), LV—left ventricular, MI—myocardial infarction, PCI—percutaneous coronary intervention, PVD—peripheral vascular (arterial) disease, SD—standard deviation, s/p—status post (history of), STEMI—ST-segment elevation myocardial infarction.

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
