# Peer review of "Changes over Time in Hemoglobin A1C (HbA1C) Levels Predict Long-Term Survival Following Acute Myocardial Infarction among Patients with Diabetes Mellitus"

_jcm, 2021, doi:10.3390/jcm10153232_

Round 1
Reviewer 1 Report
Dear authors,
The diabetes field is moving away from the historic reliance on surrogate markers, such as HbA1C; the way in which glucose levels are reduced matters for the ultimate outcome on diabetic patients and so reliance on data with respect to glycemic control efficacy is not sufficient. (JAMA 2017; 317 (10) 1017-1018 and K J Lipska).
Trials that use outcomes based solely on glycemic parameters are no longer acceptable for clinical decision making (2021 Eur Cardiol Tariq Ahmad et al., 2021 Eur Cardiol Martinez F. et al.)
Old age does not decrease HbA1c, on the contrary there is 0.1% increase in HbA1c for every 10 years (Dubowitz N et al, Diabet Med. 2014 Aug;31(8):927-35)
Using HbA1c alone to assess glycemic control can be misleading because there are many pitfalls in HbA1c measurement, which you did not take under consideration and this is a major bias of your trial (e.g iron deficiency anemia, deficiency of Vit B12 or folic acid, severe hypertriglyceridemia, severe hyperbilirubinemia, chronic salicylate ingestion, chronic opioid ingestion, lead poisoning, acute or chronic blood loss, splenomegaly, pregnancy, Vit E ingestion, red blood cell transfusion, ribavirin and interferon alpha use, haemoglobin variants, vitamin C ingestion, racial and ethnic differences, uremia, hemodialysis and EPO injections; Kaiafa et al. 2020 Postgraduate Med)
Management of hyperglycemia in acute phase of diseases does not depend on HbA1c or Fasting glucose or OGTT but on CGM (J Endocrinol Invest 2016; C Savopoulos et al., Angiology 2018 Mustafa Karakurt et al.)
Reviewer 2 Report
This reviewer has evaluated the report entitled “Changes over Time in Hemoglobin A1C (HbA1C) Levels Predict Long-term Survival Following Acute Myocardial Infarction Among Patients with Diabetes Mellitus”, by authors Ygal Plakht, Harel Gilutz, and Arthur Shiyovich.
Overall, the report is interesting and well written, and this reviewer would like to recommend acceptance after the following suggested changes are made in it:
Minor - mean and standard deviation, Student's t-test were used – the authors have probably checked for the normality of the data analyzed with these methods; could the authors clarity this point.
Minor – please check line 255 in the Discussion.
Major – most results are presented as adjusted odds ratios. The authors state that the present report “cannot show causality”, an observation with which this reviewer agrees. Data adjustment probably aims at getting nearer to a causality reasoning, but the problem is that each group of authors uses a given set of adjusting variables, making the results less easy to compare, unlike unadjusted data, which are straightforward to compare.
In short, readers might be interested in looking both at raw and at statistically manipulated data. This reviewer would therefore like to suggest a change in the text, in order to present both adjusted and unadjusted data.
Figure 1 of the Appendix (Figure A1) would appear to be the only one dealing with unadjusted data; it is easy to interpret; it appears to show a major finding of this report, since it sends readers such a powerful message.
The abstract could be rewritten, mentioning the finding of Figure A1 and the findings with unadjusted data.
Reviewer 3 Report
With this study, the authors aimed to evaluate the prognostic significance of HbA1C levels and changes among diabetic patients after non-fatal AMI.
They found that a rapid increase in HbA1C was associated with a greater risk for mortality and also that HbA1C values and their changes are significant prognostic markers for long-term mortality among AMI-DM patients. They conclude that ∆HbA1C and its timing, in addition to absolute HbA1C values, should be monitored.
In my opinion, the article is well written, easy to read, with clearly presented methods, and a very well-characterized population.
I have a few comments for the authors:
1 - Can the authors explain the relatively low rate of PCI in the population? When compared to other countries the rate is relatively low
2 - It would be relevant to know which cardiovascular drugs the patients were taking after hospital discharge
3 - In the time of new oral antidiabetic drugs, can the authors also show which antidiabetic drugs patients were taking after discharge?
4 - It would be interesting to present a subgroup analysis of patients with HbA1c < 5.5% in order to understand why these patients had a worse prognosis.
Round 2
Reviewer 1 Report
I am happy with the revised manuscript